# OpenReview forum: "Reducing Tool Hallucination via Reliability Alignment"
_ICML.cc/2025/Conference — ICML 2025 poster_

### Official Review · Reviewer_D96Q · 2025-03-12

**Overall Recommendation:** 2

**Summary:**

This paper discusses how LLMs suffer tool hallucinations, which can cause task failures and higher costs. It defines these errors as two main types: picking the wrong tool or using a tool incorrectly. The paper introduces RelyToolBench, a set of specialized tests and new metrics to measure and reduce such issues. The paper introduces Relign, a framework that helps LLMs make better tool-related decisions by allowing them to pause, ask for clarification, or switch tools when unsure. The experiments show that Relign effectively decreases tool hallucinations, making LLMs more reliable and efficient in using tools.

**Claims And Evidence:**

yes

**Essential References Not Discussed:**

None

**Experimental Designs Or Analyses:**

The main result uses specific training data sizes and model scales, which might not generalize to all LLMs or tool-use scenarios. Please show detailed results in multi scales.

**Methods And Evaluation Criteria:**

yes

**Other Comments Or Suggestions:**

See Questions

**Other Strengths And Weaknesses:**

Strengths:

The paper presents a novel framework for addressing tool hallucinations in LLMs. The categorization of tool hallucinations into selection and usage errors, with further subtypes, is a good approach to systematically addressing this issue.

Weaknesses:

While the framework is novel, it builds on existing ideas from reinforcement learning and preference optimization. The paper could benefit from a more detailed comparison with related methods/benchmarks to highlight the specific advancements it makes.

**Questions For Authors:**

1. The concept of "indecisive actions" in the Relign framework seems crucial for reducing tool hallucinations. Could you clarify the specific mechanism or threshold the model uses to decide between executing a tool call, deferring action, or seeking clarification?

2. The evaluation metrics are central to demonstrating the effectiveness of Relign. Beyond the limited human evaluation mentioned in Appendix A, have these metrics been validated against more extensive human judgments of task success and system reliability?

3. The paper shows that larger models generally perform better and hallucinate less. Have you explored how different model architectures (beyond just scale or model type(company)) might affect hallucination rates? Any insightful analysis??

**Relation To Broader Scientific Literature:**

The contributions of this paper build on and extend existing works by providing detailed evaluation metrics, a specialized benchmark, and a new alignment framework. These advancements aim to enhance the reliability LLMs in their interactions with external tools.

**Theoretical Claims:**

yes

---

> ### Author Rebuttal · Authors · 2025-04-01
>
> Thanks for your detailed review and valuable feedback. We provide our responses below:
>
> # Response about Weaknesses
> Thank you for recognizing the novelty of our framework. We clarify our key contributions as follows:
>
> ### (1) A systematic evaluation framework for tool hallucinations.
>
> Our framework integrates rule-based and LLM-based evaluations. While rule-based methods ensure syntactic correctness in tool invocation, they overlook whether **a valid invocation is actually non-hallucinatory**. This issue has been largely ignored in prior work, which often neglects tool content hallucination. Additionally, we introduce the **Reliable Pass Rate (RePR)** metric to address limitations in traditional pass rate evaluations by explicitly considering tool hallucinations.
>
> ### (2) Joint alignment of helpfulness and reliability.
>
> Unlike previous work that focuses solely on task completion, our Relign framework also emphasizes **the reliability of tool usage**. This reduces both hallucinated and ineffective tool calls while maintaining high task success rates. Our approach jointly optimizes these factors using a comprehensive **utility** metric to ensure balanced performance.
>
> ### (3) Differences from conventional SFT+DPO training.
>
> While our training builds on preference optimization, we redefine the optimization objectives by **introducing an indecisive action space**, teaching the model when to execute indecisive actions instead of hallucinating. Unlike general preference alignment methods that optimize multiple factors, we explicitly focus on helpfulness and reliability, leveraging **structured preference data** to refine tool invocation decisions. Rather than modifying the core algorithm, our contribution lies in refining **training objectives and data synthesis**.
>
>
> # Response about Q1
>
> In our work, indecisive actions are implemented as special termination functions, such as direct termination, tool switching, and TalkToUser. When the model invokes these functions, the tool invocation process ends immediately, as the model determines the task to be infeasible. The model learns to use these functions through SFT and DPO training.
>
> - **SFT stage**: We introduce the indecisive action space by modifying tasks to be unsolvable (e.g., changing toolsets) and training the model to invoke appropriate termination functions instead of hallucinating.
>
> - **DPO stage**: The model refines its decision-making through preference learning. At each tool invocation step, multiple outputs are sampled.
>
>    - If the task is feasible, the preference order is: correct tool invocation > hallucinated tool invocation > indecisive action.
>
>    - If the task is infeasible, the order is: indecisive action > hallucinated tool invocation (as correct tool use is impossible).
>
> This data-driven approach enables the model to assess the task's feasibility and determine whether to continue tool invocation or execute an indecisive action, effectively reducing hallucinations.
>
>
> # Response about Q2
>
>
> ### (1) Is there additional human evaluation for the metrics?
>
> Our evaluation metrics are improvements upon the **Pass Rate** introduced in ToolBench, which has been validated through human evaluation in its original work. While our study introduces several new evaluation metrics, they primarily rely on tool hallucination evaluation. Thus, we conducted targeted human evaluation on the LLM-based components to ensure accuracy. Given the alignment with prior metrics, we believe this evaluation is sufficient. However, if specific aspects require further assessment, **we are open to conducting additional evaluations**.
>
>
> ### (2) Is LLM-based automatic evaluation feasible and reasonable? How do we ensure evaluation accuracy?
>
> Our evaluation is divided into two specific tasks: assessing tool-task relevance and checking parameter-user input consistency. These tasks are **simpler than full tool invocation**, as critique-based judgments are easier than generative actions. We further enhance reliability by **using specialized prompts to guide assessments** and instructing the LLM to **return “unsure” when uncertain** (see Table 3 in the appendix). This approach minimizes errors and ensures accurate evaluations. Our human validation further confirms its effectiveness.
>
>
> # Response about Q3
>
> Apologies, but we have not conducted additional exploration into model architectures. To the best of our knowledge, existing research on hallucinations has not discussed the impact of different model architectures (e.g., MoE, Mamba) on hallucinations. Our work primarily focuses on proposing an evaluation framework for tool hallucinations and improving tool hallucination mitigation based on this framework. We believe investigating the relationship between model architectures and tool hallucinations is beyond the scope of this paper.
>
> We greatly appreciate your valuable feedback and look forward to your response.

---

### Official Review · Reviewer_fZsh · 2025-03-12

**Overall Recommendation:** 2

**Summary:**

LLMs can solve diverse tasks by using external tools. Investigating tool hallucination is crucial because hallucination can cause problems that are hardly recognized. This paper introduces RelyToolBench to evaluate tool hallucinations, including four types of hallucinations. Authors also propose Relign to reduce tool hallucination by SFT and DPO. As shown in the experiments section of the paper, Relign generally outperforms other baselines. In addition, the large data size increases both tool and task hallucinations.

## update after rebuttal
I would like to acknowledge the contributions of this work in terms of both the benchmark and the proposed method. I have read other reviews and rebuttal comments. However, the scope and positioning of the paper are somewhat ambiguous. Thus, I keep my original score; please see the reasons below.

The benchmark proposed in this paper is not the first to tackle tool hallucinations, which may raise questions in terms of originality. The paper needs a detailed discussion of the pros and cons of the previous work. I am not convinced by the rebuttal comment, such as the lack of taxonomy. The previous work has proposed a well-organized benchmark in terms of in-depth and in-breadth. Also, the paper had to include the discussion before submission because the previous work is easily found by topic alone and published before ICML submission. The authors highlight the alignment and training as the one of key differences, but, more analysis and investigation are needed. Based on the conducted experiments and used hyperparameters, it is difficult to understand why the proposed method is better than others.

**Claims And Evidence:**

Please refer to the weaknesses.

**Essential References Not Discussed:**

Prior work [R1] has proposed ToolBH to evaluate the hallucination in LLMs with tools. Since one of the main contributions is the evaluation of tool hallucination, discussion and comparison are needed.

[R1] Toolbehonest: A multi-level hallucination diagnostic benchmark for tool-augmented large language models, EMNLP'24

**Experimental Designs Or Analyses:**

- The hyper-parameters used in other baselines, such as SFT, DPO, and SFT & DPO, are ambiguous. For example, does Line 325, "the learning rate was set to 1e-5", mean the learning rate is used in SFT & DPO or the proposed method? The experimental details should be clear.
- I wonder if the training steps of SFT, DPO, SFT & DPO, and Relign are the same or not.
- If authors use the grid search for finding hyper-parameters, the details should be included.
- Relign is the sequential training of SFT and DPO. It is unclear why the proposed method outperforms single training of SFT and DPO and the joint training of SFT & DPO.
- In Lines 381-382, the authors said that "This could be attributed to overfitting". However, considering (1) the number of tool calling number is reduced as the data size increases and (2) the hallucination is reduced as the model size increases, the overfitting might not cause the degradation of performance. Including the additional analysis can make the claim clear.
- The authors discussed "a noticeable decline in both Tool and Task Hallucination rates" in the discussion of "Does a Larger Model Reduce Hallucination?". However, task hallucination increases as the model size increases.

**Methods And Evaluation Criteria:**

- The proposed method, Relign, is reasonable to mitigate the tool hallucination. SFT and DPO are widely used algorithms.
- The evaluation criteria used in the paper are tool hallucination rate, task hallucination rate, RePR, and Utility. The used criteria are clear.
	- Tool hallucination rate considers the number of hallucination tool calls.
	- Task hallucination rate considers the sample-level hallucination.
	- RePR is the pass rate without hallucination.
	- Utility measures the quality in terms of success, the number of calls, and hallucinations.

**Other Comments Or Suggestions:**

Please see weaknesses.

**Other Strengths And Weaknesses:**

**Strength**
- Hallucination is an important research topic to improve the responsibility.
- The paper categorizes the types of tool hallucinations and defines the metrics to evaluate the hallucination.
- Sequential training with SFT and DPO shows the higher performance than others.

**Weakness**
- The major concerns regarding experiments are mentioned in **Experimental Designs Or Analyses**, such as experiment details and analysis. Please see **Experimental Designs Or Analyses**.
- I respectfully disagree with "pioneer a comprehensive evaluation framework for Tool Reliability" because prior work [R1] has proposed ToolBH to evaluate the hallucination in LLMs with tools. Since one of the main contributions is the evaluation of tool hallucination, discussion and comparison are needed, as mentioned in **Essential References Not Discussed**.
- Does Relign also outperform others on the prior benchmarks? Reducing hallucination is important, but the performance on the existing benchmarks is questionable.

[R1] Toolbehonest: A multi-level hallucination diagnostic benchmark for tool-augmented large language models, EMNLP'24

**Questions For Authors:**

- Why does the paper not report a naive pass rate?

**Relation To Broader Scientific Literature:**

The paper is related to large language models with tools. The key contribution is the investigation of tool hallucination. It is related to the finding of prior work that LLMs have hallucinations.

**Theoretical Claims:**

There are no theoretical claims in the paper.

---

> ### Author Rebuttal · Authors · 2025-04-01
>
> Thanks for your detailed review and valuable feedback. We provide our responses below:
>
> # Re. to Experimental Designs or Analyses
>
> - (1) All experiments share the same hyperparameters unless stated otherwise. In all methods, the learning rate is set to 1e-5 for SFT and 1e-5 for DPO to ensure consistency.
>
> - (2) The SFT dataset contains 10,000 fixed samples, so the number of training steps remains the same. However, DPO training steps vary across methods and models. We select 10,000 tasks and allow multi-round interactions with the environment. In each round, we sample ten outputs at a temperature of 0.7 to construct the DPO dataset. This results in a theoretical maximum of 10,000 tasks × number of interaction rounds, but not all rounds yield valid DPO samples. In practice, the number of DPO preference pairs ranges from 10,000 to 20,000. For example, in Relign training, Toolllama, Llama3.1, and Qwen2.5 yield 17,000, 19,000, and 14,000 pairs, respectively. The final training steps depend on the number of DPO pairs, using a batch size of 32 for two epochs.
>
>
> - (3) We did not perform a grid search but used standard values (e.g., 1e-5 for SFT, 1e-6 for DPO) to ensure stable fine-tuning, focusing on training data synthesis rather than hyperparameter optimization.
>
> - (4) Relign follows a two-stage training: SFT first, then DPO on the SFT model. The key distinction between Relign and other methods lies in the base model used for DPO data sampling. In Relign, the DPO training data is sampled from a model that has undergone SFT training, whereas in other methods, it is not. We believe SFT introduces an indecisive action space, which is then refined during DPO. Without SFT, the model generates fewer indecisive actions, leading to suboptimal DPO training. This explains Relign’s superior performance.
>
> - (5) We apologize for any confusion caused. Larger models generally reduce tool hallucinations due to improved general capabilities. However, increasing training data slightly raises hallucinations, likely due to overfitting on ToolBench data, which consists of well-structured tool interaction trajectories and lacks samples that handle edge cases (e.g., failure scenarios where a task cannot be completed). Thus the model learns correct tool calls but struggles with failure scenarios, defaulting to excessive hallucinated tool calls instead of invoking indecisive actions—precisely the issue Relign aims to address.
>
> - (6) We apologize for the typo. Our intended meaning was that larger models significantly reduce both tool hallucinations and excessive tool calls, leading to more reliable and efficient tool use.
>
> # Re. to Weakness #2
> Thank you for the references. We highlight the key differences below:
>
> ### (1) Systematic Definition and Real-World Evaluation
> The cited work lacks a **comprehensive taxonomy** of tool hallucinations and **omits key types**, such as excessive tool calls (tool timing hallucinations) and fabricated tool parameters (tool content hallucinations). Additionally, its evaluation relies on sub-tasks like classification and description, which **do not reflect real-world tool use**. A model’s competence in these tasks does not necessarily prevent hallucinations during actual tool interactions. In contrast, our evaluation occurs within real-world tool-use scenarios, ensuring direct assessment of hallucinations in execution. We also introduce an **LLM-based automated evaluation** to systematically assess these hallucination types.
>
> ### (2) Beyond Evaluation: Alignment and Training
> While the cited work focuses on dataset construction, **it does not address model training or alignment**. Our work goes further by proposing a framework and data synthesis approach that effectively reduces tool hallucinations and improves tool-use efficiency and reliability.
>
> # Re. to Weakness #3
> As shown in the table below, we also evaluate our method on the out-of-domain test set APIBench. We follow the evaluation setup described in [1] (for more details, please refer to [1]). Rather than training on APIBench, we treat each API in the prompt as a function call to assess how well our trained model generalizes to OOD datasets. Experimental results with two types of tool retrieval indicate that Relign improves model performance on other tool-use tasks as well, suggesting that reducing tool hallucinations helps models learn better tool-use strategies (AST represents tool-calling accuracy).
>
> |Method| HuggingFace Hallu. | HuggingFace AST | TorchHub Hallu. | TorchHub AST | Tensorhub Hallu. | Tensorhub AST|
> |-|-|-|-|-|-|-|
> |Llama3.1 + BM25 |9.7|14.3|11.7|47.8|10.5|40.3|
> | + Relign + BM25 |**6.4**|**15.7**|**7.9**|**50.1**|**5.5**|**42.5**|
> |Llama3.1 + Oracle |9.3|87.8|10.4|86.1|7.8|89.6|
> | + Relign + Oracle |**6.5**|**89.5**|**7.7**|**88.9**|**3.2**|**91.3**|
>
> [1] Toolllm: Facilitating Large Language Models To Master 16000+ Real-World Apis.
>
> # Re. to Q1
> Please see the response to Reviewer fZsh’s Existing Metrics for more details.

---

### Official Review · Reviewer_ysVs · 2025-03-14

**Overall Recommendation:** 4

**Summary:**

This paper addresses tool hallucination in LLMs, where models incorrectly select or misuse tools. It introduces RelyToolBench for evaluation and Relign, a reliability alignment framework that enables LLMs to defer, clarify, or adjust tool use. Using SFT and DPO, Relign reduces hallucinations, improves task reliability, and enhances efficiency. Experiments show Relign outperforms baselines and achieves competitive performance with GPT-4o while reducing unnecessary tool calls.

## update after rebuttal
I’m convinced by the authors' opinion for including LLMs in the core framework. However, I don’t agree that Tool Num can serve as a sufficient proxy for efficiency. I also agree with the rest of the authors’ responses and will raise my score by one point.

**Claims And Evidence:**

It is an obvious fact that LLMs can use tools, and the associated issues are well known. This paper categorizes these issues in detail, proposes evaluation metrics, and suggests improvements using SFT and DPO.

**Essential References Not Discussed:**

If there had been related works, I would have referred to them to explore recent trends, but unfortunately, there were none.

**Experimental Designs Or Analyses:**

All evaluation metrics (RePR, Tool Hallu, Tool Num, Utility) were newly proposed, which raises the question of whether none of the existing metrics were useful.

**Methods And Evaluation Criteria:**

A significant portion of the evaluation process relies on the judgment of the LLM itself, which makes it somewhat contradictory to assess LLM hallucinations using another LLM.

**Other Comments Or Suggestions:**

The figures and experimental results were well-organized, making them easy to understand.

**Other Strengths And Weaknesses:**

I like papers that define new problems and develop benchmarks or evaluation metrics for them. However, the problem itself seems to have already been widely discussed. Additionally, the introduction of the Indecisive Action Space is likely to reduce time or computational efficiency, but there is no analysis of this aspect.

**Questions For Authors:**

1. The repeated use of a tool was portrayed negatively, but aren’t there fields where repeated use is necessary to improve accuracy?
2. In the Indecisive Action Space, how is it verified that the newly selected tool is actually correct when the tool is switched?
3. Asking the user for input is not essentially an automated methodology, is it?
4. As mentioned earlier, isn’t there an excessive reliance on LLMs to prevent incorrect generations by LLMs?

**Relation To Broader Scientific Literature:**

There would be various tools that can be used by LLMs in the neat future (space travel, airplane, military, etc.).

**Theoretical Claims:**

There are no theoretically advanced descriptions, and no major issues seem to be present.

---

> ### Author Rebuttal · Authors · 2025-04-01
>
> Thanks for your detailed review and valuable feedback. We provide our responses below:
>
> # Re. to Evaluation Criteria & Q4
>
> ### (1) Why did we introduce LLM-based evaluation?
>
> Our evaluation process for tool hallucinations consists of both rule-based evaluation and LLM-based evaluation. Rule-based evaluation can determine whether the LLM’s tool calls follow correct syntax. However, a key question arises: **Is a syntactically valid invocation actually non-hallucinatory?** This is why we introduced LLM-based evaluation—only an LLM can assess aspects beyond syntactic correctness and evaluate the content itself. LLM-based evaluation has also been widely used in prior research on hallucination assessment [1-2].
>
> ### (2)How do we ensure evaluation accuracy?
>
> In our LLM-based evaluation process, we decompose the assessment into specific evaluation tasks. On the one hand, **these evaluation tasks are significantly simpler than the full tool-calling task**, as making a critical judgment is easier than generating a valid action. On the other hand, we design **dedicated evaluation prompts** tailored to each evaluation task, guiding the LLM to assess various aspects in a structured manner. Furthermore, we explicitly **instruct the LLM to output unsure when it is uncertain**, rather than forcing it to provide a definitive evaluation result (as shown in Table 3 in the appendix).
>
> In summary, we believe that using LLMs for automated evaluation is reasonable, and our human evaluation results further validate its accuracy.
>
> [1] FACTSCORE: Fine-grained Atomic Evaluation of Factual Precision in Long Form Text Generation.
>
> [2] SelfCheckGPT: Zero-resource black-box hallucination detection for generative large language models.
>
> # Re. to Computational Efficiency
> While we did not conduct direct tests on the actual computational efficiency or time reduction, we actually use **Tool Num as an indicator for computational efficiency**. Since the model calls a tool once per interaction round, the average number of tool calls effectively corresponds to inference time and output length. Compared to dynamic and less stable metrics, the number of tool calls remains relatively fixed. Therefore, we primarily use the Tool Num metric to reflect the efficiency gains achieved by our method. As shown in Table 2, our model significantly reduces the average number of tool calls per task (ToolLlama reduces from 3.7 to 0.9, Llama3.1 from 2.2 to 1.5, and Qwen2.5 from 2.4 to 1.7).
>
>
> # Re. to Existing Metrics
>
> Our RePR metric is a refined version of Pass Rate, as we found that some tasks in the original pass rate metric exhibited result hallucinations. Therefore, we believe the pass rate has certain inaccuracies and did not include its results in our paper. As shown in the table below, we have supplemented some of the original Pass Rate results. The results indicate that RePR is lower than Pass Rate, and the gap is more pronounced in base models without Relign alignment. This suggests that unaligned models tend to produce more tool hallucinations, which in turn mislead the final results. Moreover, whether using Pass Rate or RePR, our Relign framework consistently improves task success rates. We will include the full results table in future versions of our paper for reference.
>
> |Method|I1  PR/RePR|I2 PR/RePR|I3 PR/RePR| Overall PR/RePR|
> |-|-|-|-|-|
> |Toolllama |76.8/64.2|77.0/64.4|72.3/58.9|75.4/62.5|
> | + Relign |77.1/68.2|77.3/68.3|76.2/71.0|76.9/69.1|
> |Llama3.1 |83.0/68.5|84.8/66.1|80.1/61.3|82.6/65.3|
> | + Relign |87.0/80.5|89.2/75.8|86.9/75.4|87.7/77.2|
> |Qwen2.5 |86.3/71.5|84.3/66.2|80.6/57.7|83.7/65.1|
> | + Relign |87.6/77.0|87.5/69.0|85.5/72.9|86.9/73.0|
>
>
> # Re. to Q1
> In some special cases, repeated tool calls may be beneficial (e.g., when a tool fails due to high latency). However, our evaluation environment is based on StableToolBench, which **ensures that the tool environment remains stable**. In this scenario, repeated tool calls are typically an **abnormal behavior** of the model. Besides, in our specific evaluation of repeated tool calls, a tool call is only considered a hallucination if both the tool call itself and its environmental feedback are identical. This approach can partially address your concern.
>
> # Re. to Q2 and Q3
> In the Indecisive Action Space, ChangeTool and TalkToUser are designed as **two special termination functions**. When the model invokes either of these functions, the tool invocation process is immediately terminated, as the model has determined that the task cannot be completed in the current environment. As a result, we did not design specific pipelines for tool switching (which would require a tool retriever ) or for user interaction (which would require a user simulator). Instead, we directly exit the process upon encountering these exceptional actions. We will explore how to better design task execution workflows following exceptional actions in future work.

---

### Official Review · Reviewer_AGyu · 2025-03-17

**Overall Recommendation:** 3

**Summary:**

The paper focuses on the problem of tool hallucinations. The authors categorize tool hallucinations into selection and usage hallucinations, providing a framework for targeted improvements. A new benchmark is introduced to capture these categories (RelyToolBench) and model fine-tuning methods (Relign) are explored to fix these hallucinations.

**Claims And Evidence:**

The paper dives deeper into source of hallucinations in language model outputs when using tools. They find that models can not only generate invalid tool invocations but also not realize how many or which tools to use. The authors validate these on RelyToolBench and have discussed the construction of RelyToolBench. A concern here is that RelyToolBench is artificially constructed by purposely introducing errors, such as partial tool information in the prompt, and performs evaluations on a distorted distribution. The real-world generalization of the findings are unclear.

The new training method proposed “Relign” is the typical way of performing post-training of large language models [1, 2], i.e., SFT followed by DPO. They report a baseline (SFT&DPO) which is not as widely adopted a strategy. So the proposed method is not really new as such, and the Relign method has not been discussed as such.

[1] Grattafiori, Aaron, et al. "The llama 3 herd of models." *arXiv preprint arXiv:2407.21783* (2024).

[2] Mehta, Sachin, et al. "Openelm: An efficient language model family with open training and inference framework." *Workshop on Efficient Systems for Foundation Models II@ ICML2024*. 2024.

**Essential References Not Discussed:**

There is a large body of work concerning hallucinations and RAG which has not been adequately discussed in this paper. I think RAG serves as a simple baseline to reduce tool hallucinations and I did not follow the arguments to perform the RAG experiment for this paper. If the RAG experiment is not possible, at least the paper should recommend the reader about when to perform RAG and when to use their method.

**Experimental Designs Or Analyses:**

1. The paper identified hallucination issues by creating a new benchmark. This benchmark is synthetically generated, which is largely OK, but some concerns raised above need to be addressed.
2. See comments on RAG under various headings below.

**Methods And Evaluation Criteria:**

1. RePR = Pass rate - task hallucination rate. If task hallucination occurs, I assume that pass rate will be also be zero in that scenario. So, the model is penalized multiple times for failures here. I’m not sure if it makes sense to combine these into a single metric. Why not just view these separately, as they are?
2. SFT&DPO is not a required baseline. Relign should be identified as SFT+DPO with the relevant references identified.

**Other Comments Or Suggestions:**

N/A

**Other Strengths And Weaknesses:**

The main novelty in the paper arises from the way the data is constructed and analyses around sources of hallucinations. These would serve as meaningful insights to the community.

**Questions For Authors:**

1. How does RAG compare with the methods discussed to improve the hallucination rates? I understand a paper cannot have all experiments but I think RAG is simple and popular enough to concretely justify the need for post-training over RAG to reduce hallucinations. There are pros/cons of each and I think the paper should highlight these for the application(s) being discussed. When should someone follow the research of this work and not RAG methods for their work? Questions like these remain unanswered in the paper.
2. RelyToolBench and the preference datasets were synthetically created. How do you justify the use of the training data and the evaluations to reflect real-world scenarios? Can you comment on this distribution shift?

**Relation To Broader Scientific Literature:**

The paper is related to the body of work on tool usage with LLMs. It is also related to post-training strategies to reduce hallucinations. The paper dives deep into the mode of hallucinations [1, 2] for tool [3, 4] usage using a new benchmark and applies various post-training strategies [5, 6] to reduce hallucinations.

[1] Varshney, Neeraj, et al. "A stitch in time saves nine: Detecting and mitigating hallucinations of llms by validating low-confidence generation." arXiv preprint arXiv:2307.03987 (2023).

[2] Jain, Nihal, et al. "On Mitigating Code LLM Hallucinations with API Documentation." arXiv preprint arXiv:2407.09726 (2024).

[3] Patil, Shishir G., et al. "Gorilla: Large language model connected with massive apis." Advances in Neural Information Processing Systems 37 (2024): 126544-126565.

[4] Schick, Timo, et al. "Toolformer: Language models can teach themselves to use tools." *Advances in Neural Information Processing Systems* 36 (2023): 68539-68551.

[5] Grattafiori, Aaron, et al. "The llama 3 herd of models." *arXiv preprint arXiv:2407.21783* (2024).

[6] Mehta, Sachin, et al. "Openelm: An efficient language model family with open training and inference framework." *Workshop on Efficient Systems for Foundation Models II@ ICML2024*. 2024.

**Theoretical Claims:**

N.A

---

> ### Author Rebuttal · Authors · 2025-04-01
>
> Thanks for your detailed review and valuable feedback. We provide our responses below:
>
> # Response about RePR
> We apologize for any confusion caused by the description of the metric. In fact, **the model is not penalized multiple times**. RePR represents the proportion of tasks that were originally considered as successfully passed but were later verified to be free of result hallucinations through our hallucination detection. The deducted portion consists of tasks that were initially deemed successful but were later identified as containing result hallucinations. Importantly, tasks with a pass rate of 0 are not considered as having result hallucinations.
>
> # Response about RAG and Q1
> ### (1) Relevance of Our Work to RAG
> In our tool-based tasks, we provide the model with relevant information about each tool through the api description field in the prompts, as shown in Table 7 in the appendix. This includes descriptions of tool functionalities, tool parameters, and example values of parameters. **This setup can be considered a special form of RAG tailored to tool documentation.**
>
> Since the APIs in our dataset do not appear in the model’s training data, feeding tool API documentation into the model is essential; otherwise, the model would not be able to complete the tasks.
>
> ### (2) Why RAG is Not Suitable for Tool Hallucination Scenarios
> As mentioned above, we already provide the model with tool API usage instructions in the prompt, yet the model still exhibits different types of tool hallucinations during task execution. This could be due to the relatively **limited knowledge** contained in RAG-based knowledge retrieval.
>
> Another possible way to construct a RAG knowledge base is to generate and retrieve real usage examples for specific tool APIs, similar to in-context learning. However, in our inference setting, many of the APIs we encounter are not present in the training database. This makes it difficult to pre-prepare corresponding demonstration examples. A potential alternative would be to synthetically generate a large number of tool invocation examples and then filter out high-quality examples. However, this approach does not generalize well to large-scale real-world tool-use scenarios and is beyond the scope of this paper.
>
> ### (3) Discussion on RAG and Post-Training for Addressing Hallucination
>
> **a. Challenges in Constructing RAG Knowledge Bases**
>
> RAG relies on external knowledge bases, which can be difficult to construct. If the knowledge base from the training data does not sufficiently cover the inference-time data, applying RAG becomes challenging.
>
> **b. Fundamental Differences Between RAG and Post-Training**
>
> RAG is designed to mitigate hallucinations **by introducing external knowledge at inference time**. Unlike post-training methods, RAG does not update the model’s parameters, making it relatively lightweight. However, RAG typically retrieves specific pieces of information, which at most allows the model to mimic certain patterns, similar to in-context learning.
>
> In contrast, post-training methods can deeply **alter the model’s cognition and capabilities through training**. When hallucinations stem from the model’s inherent weaknesses in understanding and interacting with the tool environment, RAG combined with prompting may be insufficient to mitigate these hallucinations. This is particularly relevant for the tool hallucinations observed in our experiments. In such cases, post-training is necessary to enable the model to better learn and model the tool interaction environment.
>
> # Response to Q2
> First, the RelyToolBench dataset includes both the original subset from StableToolBench and additional subsets specifically designed to evaluate particular hallucinations, namely the Unmatched Tools and Miss Parameter subsets. It is important to emphasize that the **original subset closely resembles real-world applications**. However, it also contains four different types of hallucinations as shown in Figure 6 in our paer, and both its tasks and APIs are relatively realistic. This demonstrates that **similar hallucinations occur in real-world scenarios**, not just in our two synthetically constructed subsets.
>
> Additionally, as described in our response to Reviewer fZsh’s Weakness#3, we conducted **evaluations on an out-of-distribution test set APIBench**. We observed that models trained with our Relign method did not show a decline in tool-calling capabilities; in fact, they even exhibited some improvements. This provides further evidence **supporting the generalization ability** of our approach.
>
> # Response to Scientific Literature
>
> Thank you for providing the references. We will cite them in future versions of our paper and supplement our discussion accordingly.
>
> We greatly appreciate your thoughtful feedback and look forward to your response.

---

### Decision · Program_Chairs · 2025-05-01

**Decision:**

Accept (poster)

**Comment:**

This paper addresses the problem of LLM's using the wrong tool or hallucinating tool parameters by introducing RelyToolBench, a new benchmark that extends StableToolBench with two new subsets and new metrics. Further, the authors propose Realign, a finetuning approach using synthetic data, along with SFT and DPO, to enable an LLM to better select tools or seek clarification from the user.

The paper received 4 reviews with split scores: weak accept, accept, weak reject, weak reject. All reviewers acknowledged the authors rebuttal.

The most substantive concerns raised by reviewers include:
1. Concerns about the benchmark being synthetically generated, and may not generalize. In response, the authors provided an evaluation on the out-of-distribution test set from APIBench, showing improved performance over baseline Llama3.1 setups.
2. Lack of novelty in 'Realign'. The AC agrees that effectively, Realign is SFT + DPO, and the main contribution to the method is including the examples of seeking clarification in the training data.
3. Missing connections to previous work on hallucinations, and in particular the paper Toolbehonest: A multi-level hallucination diagnostic benchmark for tool-augmented large language models, EMNLP'24 (published several months before the ICML deadline). The AC's opinion is that missing ToolBH is a significant oversight. In rebuttal, the authors argued to diminish the contributions of ToolBH.
4. Metrics are new and unproven. Here, the AC agrees with the authors that the new metrics are simple and intuitive enough that verification (for example, via correlation with human judgements) would not be required.

Overall, the main new idea in the paper is training the LLM to seek clarification rather than hallucinate a tool call. The AC read ToolBH and confirmed that this contribution is novel. Therefore, the paper warrants acceptance provided the authors can update the related work to include the missing references (with a gracious assessment of the value of previous work).